# Preliminary Studies on the Intrahepatic Anatomy of the Venous Vasculature in Cats

**DOI:** 10.3390/vetsci9110607

**Published:** 2022-11-02

**Authors:** Mélanie Davy Metzger, Elke Van der Vekens, Juliane Rieger, Franck Forterre, Simona Vincenti

**Affiliations:** 1Division of Small Animal Clinical Surgery, Vetsuisse-Faculty, University of Bern, 3012 Bern, Switzerland; 2Division of Clinical Radiology, Vetsuisse-Faculty, University of Bern, 3012 Bern, Switzerland; 3Department of Veterinary Anatomy, Vetsuisse-Faculty, University of Bern, 3012 Bern, Switzerland; 4Department of Human Medicine, Faculty of Medicine, MSB Medical School Berlin, 14197 Berlin, Germany

**Keywords:** anatomy, cat, hepatic surgery, venous vasculature

## Abstract

**Simple Summary:**

The anatomy of the intrahepatic veins in cats has never been thoroughly described; thus, veterinary surgeons have based their techniques on previous knowledge about canine liver anatomy for hepatic surgeries in both dogs and cats. We used corrosion cast techniques and advanced imaging modalities on feline cadavers to describe the anatomy of the portal and hepatic veins in the feline liver. The anatomy seems consistent with that in previous studies in dogs; nevertheless, several relevant vascular differences could be identified between specimens and species and should be assessed pre-operatively to avoid surgical complications.

**Abstract:**

Hepatic surgeries are often performed in cats to obtain a disease diagnosis, for the removal of masses, or for the treatment of shunts. Whereas the vascular anatomy of the liver has been studied in dogs, such evidence is lacking in cats. The current study used corrosion casts of portal and hepatic veins and computed tomography (CT) analysis of the casts to identify and describe the intrahepatic anatomy in healthy cat livers (*n* = 7). The results showed that feline livers had a consistent intrahepatic portal and venous anatomy, with only minor disparities in the numbers of secondary and tertiary branches. The feline portal vein consistently divided into two major branches and not three, as previously described in the literature for cats. The finding of a portal vein originating from the right medial lobe branch leading to the quadrate lobe in 4/7 specimens is a novelty of the feline anatomy that was not previously described in dogs. Partial to complete fusion of the caudate process of the caudate and the right lateral lobe, with a lack of clear venous separation between the lobes, was present in two specimens. These findings allowed a detailed description of the most common intrahepatic venous patterns in cats. Further anatomical studies should be encouraged to confirm the present findings and to investigate the utility of this information in surgical settings.

## 1. Introduction

Hepatic surgery in cats is mainly performed to obtain a diagnosis and for the removal of masses, such as benign or malignant neoplasia, parasitic cysts, or abscesses; in these cases, surgical removal using partial to complete lobectomy remains the treatment of choice [1]. The liver is frequently involved in metastasis of abdominal neoplasia due to the migration of neoplastic cells along the portal system; primary malignant tumors of the liver and bile ducts, such as hepatocellular carcinoma and bile duct adenomas, have also been described in cats [2,3,4]. A good anatomical knowledge of the venous inflow and outflow of the hepatic lobes is of particular relevance when the aim of surgical excision is a curative-intent surgery, as the tumor cells are most likely to metastasize along the main vessels, as well as in the directly adjacent tissues. Furthermore, hemorrhage from the main hepatic vessels is a major complication that is directly related to hepatic surgeries [1,5]. Hemostasis can be achieved through direct ligation of the main veins after blunt finger or needle dissection, the guillotine technique, surgical stapling, and electrosurgical vessel sealant [1,5,6]. Another rare but challenging pathology in the feline liver is the presence of an intrahepatic portosystemic shunt. To be able to correctly identify the feeding portal tributary vein and the draining hepatic vein, a solid knowledge of the vascular anatomy is paramount [1,5,7].

The absence of well-delineated fissures in the human liver justified an in-depth description and definition of functional segments according to the vasculature [8,9,10]. The use of advanced corrosion casting techniques has allowed an accurate description of each liver segment and its tributary vessels, thus allowing more precise planning of surgical procedures. Whereas the vascular anatomy of the liver has been studied in dogs, such evidence is lacking in cats [11,12,13,14,15,16]. The exact venous anatomy of the feline liver remains unclear, and the techniques used for surgical hepatic procedures, such as partial and complete lobectomies, rely mainly on more detailed descriptions of canine anatomy.

The use of corrosion cast studies allows direct macroscopic visualization of liver veins. The concomitant use of more advanced imaging (computed tomography (CT)) facilitates a more accurate and precise description of the vasculature with the help of 3D-reconstruction software. Therefore, this study aimed to identify and describe the intrahepatic vascular anatomy of the feline liver, with the hypothesis of consistent anatomy, and to compare the results with the previously published knowledge of canine liver anatomy.

## 2. Materials and Methods

This study was conducted using corrosion casts of feline livers. The livers were harvested from seven European Shorthair adult cats that were more than one year of age, between 4.2 and 5.8 kg of body weight, and dead or euthanized for reasons unrelated to the study. No history of hepatic disease or laboratory and necroscopic signs of liver abnormalities were present. All owners signed a consent form in which they agreed to donate their animals’ bodies for research and teaching. The cadavers were frozen immediately after death or dissected within hours of death.

The livers were surgically resected from cadavers after thawing to room temperature for 24 h. The abdominal wall was opened using routine median celiotomy, and the main hepatic vessels were identified. The post-hepatic caudal vena cava (CdVC; Vena cava caudalis) was transected directly caudally to the heart, and the pre-hepatic CdVC cranial to the right phrenicoabdominal vein. The portal vein (PV; Vena porta) was transected between the caudal mesenteric and splenic veins. Thereafter, the splenic, right gastric, and gastroduodenal veins were ligated. 

Both hepatic inflow and outflow venous systems were catheterized with 18–22 G venous catheters and fixed using simple interrupted sutures (Ethicon Vicryl; Ethicon Inc., Raritan, NJ, USA) (Figure 1). The veins were then cast ex situ with a polyurethane acrylic resin (Synolite 0328-A-1; DSM Composite Resins AG, Schaffhausen, Switzerland), with the specimens positioned in a plastic container mimicking the liver position in a cat when placed in dorsal recumbency for surgery. The containers were filled with an isotonic saline solution (NaCl 0.9%) to decrease the pressure points between the liver lobes and quicken the polymerization process. First, the resin was mixed under a fume hood with different commercially available epoxy coloring dyes for each venous system (blue: CdVC, purple: PV). For each liver, two catheters were oriented in a standardized manner during and after the injection of resin to facilitate a later comparison of the results. The injected specimens were then left in saline-filled containers under a fume hood to allow the resin to polymerize for at least 24 h.

CT scans of the livers with the resin casts were performed the day following casting using a helical scan technique with a 1 mm slice thickness, 0.5 mm increment, 120 kV, and 140 mAs (Brilliance CT 16 slice; Philips Medical Systems, Eindhoven, the Netherlands). The livers were scanned in the same saline-filled containers to ease the distinction between the liver lobes in the scans. The acquired images were reconstructed using a bone and soft tissue algorithm before being stored in the PACS system (IMPAXEEServer_Rad, Agfa HealthCare, Mortsel, Belgium). The scans were analyzed as multiplanar reconstructed images using a DICOM reader (DeepUnity R20 XX, DH Healthcare GmbH, Bonn, Germany) to evaluate the number and location of the vascular branches of the intrahepatic PV and hepatic veins. Primary branches were defined as venous branches arising directly from the PV or joining the post-hepatic CdVC, while secondary branches were the further veins that ramified inside one lobe from the primary branches. Tertiary branches were defined as further ramifications of secondary branches within one single liver lobe. In addition, their relationships with different lobes were recorded. Three-dimensional volume-rendered images and maximum intensity projections were produced to allow easy comparison of the liver vasculature (Figure 2). The livers were macerated using an enzymatic solution (Biozym SE; Spinnrad GmbH, Bad Segeberg, Germany) at 60 °C for approximately 36 h. The remaining parenchyma was removed via gentle rinsing with water. The tertiary and smaller branches of the veins were manually removed, allowing a better macroscopic view of the main branches of the different venous systems.

The standardized nomenclature of the Nomina Anatomica Veterinaria was used to define the hepatic vessels as they were first mentioned in the results [17]. For means of readability, any further mention of the same anatomical landmark is made using the English terminology. The liver lobes were equally defined as follows: left lateral liver lobe, Lobus hepatis sinister lateralis; left medial liver lobe, Lobus hepatis sinister medialis; quadrate lobe, Lobus quadratus; papillary process, Processus papillaris; caudate process, Processus caudatus; right medial liver lobe, Lobus hepatis dexter medialis; right lateral liver lobe, Lobus hepatis dexter lateralis. 

## 3. Results

Corrosion castings were created using a standardized technique that allowed good-quality models of the intrahepatic vasculature. All casted venous systems were correctly filled with resin, except for the hepatic vein of the papillary process of the caudate lobe in one specimen, which could not be identified either on the CT scan or on the final cast; this was presumably due to poor filling of this vein during the injection.

### 3.1. Portal Venous System

In all examined specimens, the PV divided into two main branches immediately after its entrance into the liver parenchyma at the level of the hilus. The first main branch was the right PV (Ramus dexter). This rather short vein supplied the right part of the liver, giving off varying numbers of branches to the right lateral and caudate processes of the caudate lobe. The second main branch was the left PV (Ramus sinister), which was a much larger vein than the right PV and irrigated the left part of the liver (left medial and lateral, quadrate, papillary process of the caudate and right medial lobes). The first major branch was present at the level of the right medial lobe, followed by one or two smaller branches, leading to the papillary process of the caudate lobe. The main left PV then divided at its end into several branches, supplying the quadrate, left lateral, and left medial lobes (Figure 3).

### 3.2. Hepatic Venous System

The hepatic venous system could be divided into three major branches: the right, central, and left hepatic veins. The right hepatic vein (V. hepatica dextra) could also be defined as the intrahepatic CdVC crossing the caudate process of the caudate lobe to join the large post-hepatic CdVC immediately caudal to the diaphragm. Along its course through the liver parenchyma, the right hepatic vein was joined by branches coming from the right medial and right lateral lobes, as well as branches from the caudate and papillary processes of the caudate lobe. The central hepatic vein (V. hepatica media) was a medium-sized trunk formed by veins of the quadrate and right medial lobes that entered the intra-hepatic CdVC directly or joined the left or right hepatic vein shortly before their entrance into the CdVC. The veins originating from the left lateral and left medial lobes met to form the left hepatic vein (V. hepatica sinistra), which entered the intra-hepatic CdVC alongside the two other main hepatic veins (Figure 4).

### 3.3. Anatomical Variations

Some important variations in the venous systems and liver lobe anatomy were noted between the specimens. In particular, the number of tertiary veins supplying each liver lobe was subject to variation (Table 1). 

The PV branches supplying the left lateral, left medial, right medial, and papillary process of the caudate lobes were consistent with the previously described left PV. The right lateral lobe and caudate process of the caudate lobe were consistently supplied by branches of the right PV. In one specimen, an additional branch originating directly from the main PV supplied the caudate process (Figure 3(A3)). In another specimen, the right medial liver lobe was additionally supplied by a small vein originating directly from the main PV before its entrance into the liver parenchyma (Figure 3(A4)). The quadrate lobe was supplied by its branching of the left PV and by a branch originating from the vein supplying the right medial lobe in four specimens. In two specimens, the two portal veins supplying the quadrate lobe originated directly from the left PV (Figure 3(A2)). In one specimen, the quadrate lobe had two additional branches originating from the right and left medial veins (Figure 3(A5)). In two specimens, the left lateral and left medial lobes shared two common PV trunks that separated into both lobes, additionally to the secondary veins supplying both lobes separately from the left PV. Similarly, in three specimens, the right lateral lobe and caudate process of the caudate lobe had some additional shared common primary portal veins that supplied both lobes. In one specimen, the left medial and quadrate lobes shared a common portal venous primary branch.

The hepatic vein branches coming from the left lateral and left medial lobes joined the left hepatic vein, except for a vein of the left medial lobe exiting into the central hepatic vein in one specimen (Figure 4(B2)). The veins of the quadrate lobe drained into the central hepatic vein in all specimens, whereas the right medial lobe consistently gave off a secondary branch to the central hepatic vein, as well as a branch to the intra-hepatic CdVC. The hepatic veins of the papillary and caudate processes of the caudate lobe mostly drained directly into the right hepatic vein. In one specimen, no hepatic vein branch from the papillary process of the caudate lobe was identified on either the CT scan or the final corrosion cast. In three specimens, one branch of the caudate process formed a common trunk with a hepatic vein originating from the right lateral lobe (Figure 4(B3)). In the right lateral lobe, the branches of half of the specimens directly joined the right hepatic vein; in the other half, they formed a previously described common trunk with a branch of the caudate process. The specimen with a complete lobe fusion of the right lateral lobe and caudate process of the caudate lobe was excluded from this result interpretation, as the vessel separation between the lobes was not clearly defined. 

In two specimens, the right lateral lobe and caudate process of the caudate lobe were partially or completely fused. In one specimen, the venous anatomy still permitted a clear separation of the two lobes, whereas in the second specimen, the portal supply divided into two main branches, further splitting into several tertiary branches that supplied the whole fused lobe with no clear separation of the veins according to the normally described lobe anatomy. In the same specimen, five hepatic veins similarly joined the right hepatic vein without clearly forming two anatomically separated lobes (Figure 3(A6) and Figure 4(B4)).

## 4. Discussion

The results of this study describe the anatomy of the hepatic portal and venous vasculature in cats in detail, as it has not been investigated in previous studies. The anatomy seems mostly consistent with the previously described general distribution of the canine liver venous systems, while also showing some smaller variations unique to the feline liver. 

A major difference between our findings and the current literature that describes the anatomy of the intrahepatic portal venous systems in cats is the consistent finding of the PV dividing into two major branches (the right and left PV) and not three, as previously described, similarly to the course of the PV in dogs [1,5]. 

When comparing the results of this study with those of recent studies describing the portal and hepatic venous anatomy of the canine liver, a consistent basic anatomy of the intrahepatic venous systems was confirmed regarding the number of branches supplying each liver lobe, their origins, and their intrahepatic course [11,12,13,14,15,16]. A few divergences in the number of branches and course of the portal and hepatic veins of the canine livers have been reported and can also be described in cats [11,12,13,14,15,16]. In several specimens, adjacent liver lobes were supplied by a common trunk of either the hepatic or PV, as previously described in dogs [11,12,13,15,16]. For instance, in one specimen, the left medial and quadrate lobes shared a common portal venous primary branch; the right lateral lobe and caudate process of the caudate lobe shared a common hepatic venous primary branch in three specimens. This information could imply that a lobectomy of the left medial, quadrate, and right medial lobes and caudate process of the caudate lobe may compromise the blood supply of one of the adjacent lobes in the case of deep or hilar liver lobe resection. 

We observed a portal vein supplying the quadrate lobe that originated from one of the branches of the right medial lobe in four of the seven specimens. This major variation seems to be unique to the feline liver, as this variant has never been described in previous studies on canine vascular anatomy [11,12,13,14,15]. Another vascular variation that has not been described in dogs is the presence of smaller portal veins directly originating from the intra-hepatic main PV in two specimens. Overall, the number of tertiary branches of both the portal and hepatic venous systems described in the feline specimens was slightly higher than that previously described in dogs [11,12,13,14,15,16].

Finally, the partial fusion of the caudate process of the caudate lobe and the right lateral lobe in two specimens—specifically, the lack of venous separation between the inflow and outflow systems of the two lobes in one of the specimens—is another anatomical variation that has not been described previously. Further investigation of the prevalence of this lobar fusion within the feline population would be of interest.

The choice was made to perform this study on European Shorthair cats, as this breed is the most common one in Switzerland, was not developed through selective breeding, and is most likely to be the most representative in a preliminary study. Nevertheless, the low number of specimens in the study did not allow for statistical analysis, and the current results cannot yet be extrapolated to other cat breeds, as many different variations occurred in such a low number of cases.

The CT images of the casted livers allowed reliable and detailed multiplanar and 3D reconstruction of the corrosion castings without taking the risk of breaking the final corrosion casts after maceration, while the latter helped to confirm the CT observations, serving as a spatial and visual model with the help of the color-coding system. The two different venous systems showed distinct separation, and the course of the veins into the liver parenchyma could be easily followed. The use of an isotonic saline solution during CT imaging of the resin-casted livers allowed easier differentiation of the liver lobes on the dissected livers, since the physiological in vivo position was not preserved. The choice was made to use an acrylic resin for corrosion casting and imaging instead of a more traditional CT angiography with a water-soluble contrast agent to assess the intrahepatic venous systems. The high viscosity of the resin limited the vascular filling to the larger main hepatic veins, which were the objects of interest in the study. The use of a water-soluble iodine-based agent would have filled even the smaller veins and caused contrast enhancement of the liver parenchyma, in addition to increased risks of extravasation in frozen–thawed cadavers, making the interpretation of the CT imaging more challenging [18]. Furthermore, mixing variable concentrations of water-soluble contrast agent with the acrylic resin to help in the differentiation between the venous systems in imaging proved impossible due to the poor mixing ability with the chosen resin, as well as impaired resin curing after mixing. Radiographic angiography has been previously reported, but this technique has been leading to difficulties in interpretation due to the superimposition of the venous systems [19]. In vivo non-selective CT angiography with the reconstruction of the portal venous system in pigs has demonstrated high precision compared to subsequent in situ casting of the same portal venous system. Therefore, this technique seems to be a reliable method of imaging the intrahepatic vasculature and could be used in cats for precise preoperative planning of liver surgery [20].

To avoid difficulties and errors in interpreting CT images, only the two venous systems were investigated, similarly to studies conducted in dogs [11,12]. In fact, crossing or very close positions of multiple systems can reach the limits of spatial resolution, preventing accurate separation of the systems and their ramifications. In dogs and humans, the arterial and biliary systems appear to follow a consistent course along the portal veins, which requires further investigation in cats [8,14]. Microsurgery techniques for isolation and intra-abdominal catheterization of the hepatic artery and common bile duct might be helpful for further investigation of the vascular and biliary anatomy of the feline liver. Further anatomical studies using such techniques are envisaged. This study included a limited number of specimens, which was in line with previous canine studies, but further studies including a larger number of cases should be encouraged, in addition to an investigation of the arterial and biliary systems [11,12,13,14,15]. A comparison between live feline specimens with in vivo non-selective CT angiography and ex vivo post-mortem corrosion castings was not performed and could represent a further limitation of the study.

## 5. Conclusions

The location and number of the portal and hepatic venous systems were mostly consistent and similar, with different variations observed among specimens. This study allowed the first documented and detailed description of the most common intrahepatic venous patterns in cats. Further anatomical studies should be encouraged to confirm the present findings and to investigate the clinical application of the results of this study.

## Figures and Tables

**Figure 1 vetsci-09-00607-f001:**
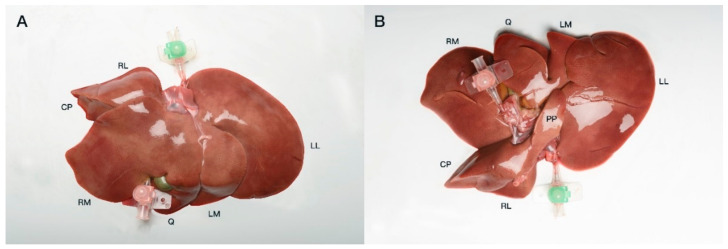
Visceral view of a feline liver prior to casting with both PV (pink catheter, caudal) and post-hepatic catheterized CdVC (green catheter, cranial) (**A**). Diaphragmatic view of the same liver (**B**). CP, caudate process caudate lobe; RM, right medial; Q, quadrate; LM, left medial; LL, left lateral; PP, papillary process caudate lobe.

**Figure 2 vetsci-09-00607-f002:**
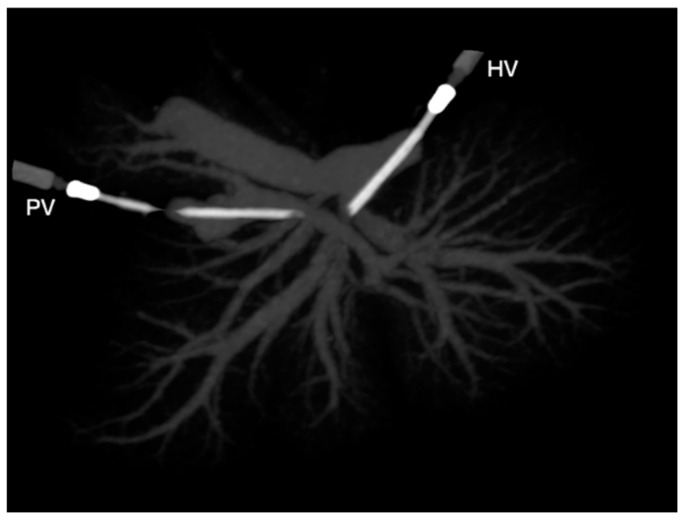
Ventrodorsal view on a maximum-intensity projection (MIP) of a liver cast. HV, catheter to the hepatic vein; PV, catheter to the portal vein.

**Figure 3 vetsci-09-00607-f003:**
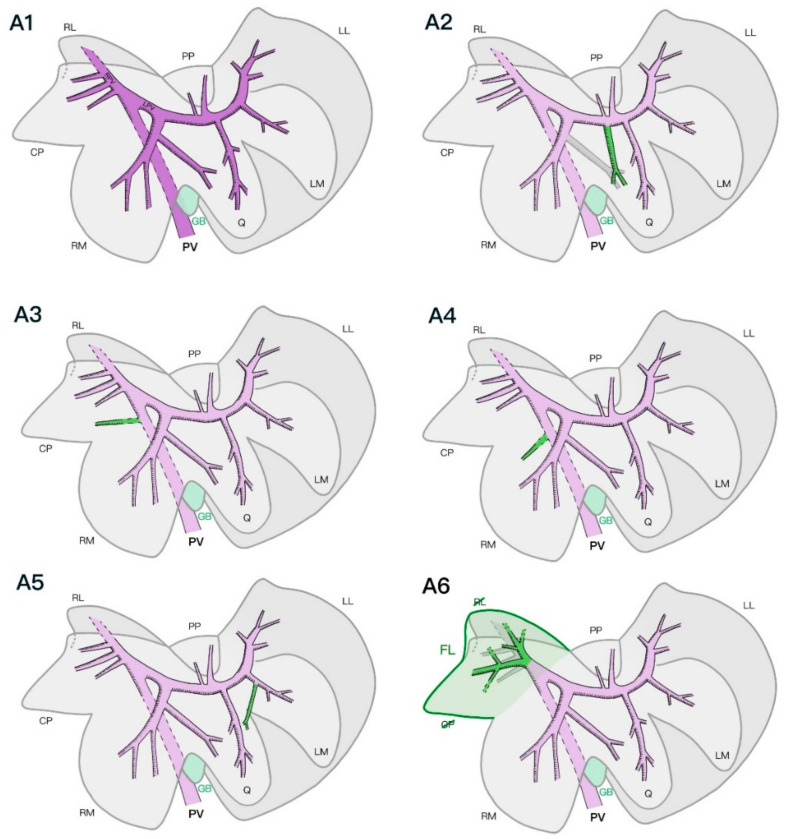
Schematic representation of the intrahepatic portal venous system with the most common number of tertiary branches and its anatomical variations from a parietal view. Most common portal pattern (**A1**), additional vein to the quadrate lobe from the left portal vein in 2/7 specimens (**A2**), additional vein to the caudate process in 1/7 specimens (**A3**), additional vein to the right medial lobe in 1/7 specimens (**A4**), additional vein to the quadrate lobe from the left medial lobe in 3/7 specimens (**A5**), and fused caudate process and right lateral lobes in 2/7 specimens (**A6**). PV, portal vein; LPV, left portal vein; RPV, right portal vein; RL, right lateral; CP, caudate process caudate lobe; RM, right medial; GB, gallbladder; Q, quadrate; LM, left medial; LL, left lateral; PP, papillary process caudate lobe; FL, fused lobe.

**Figure 4 vetsci-09-00607-f004:**
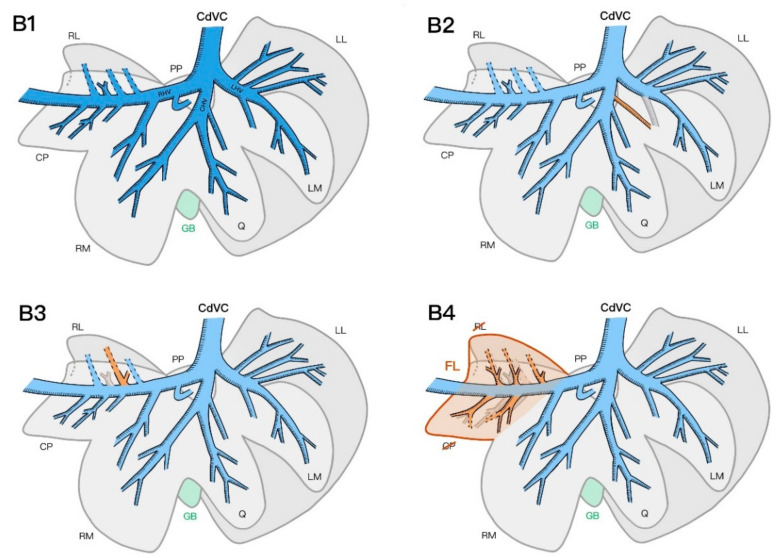
Schematic representation of the intrahepatic hepatic venous system with the most common number of tertiary branches and its anatomical variations from a parietal view. Most common hepatic venous pattern (**B1**), additional vein to the left medial lobe from the central hepatic vein in 1/7 specimens (**B2**), common trunk to the right lateral lobe and caudate process in 3/7 specimens (**B3**), and fused caudate process and right lateral lobes in 2/7 specimens (**B4**). CdVC, caudal vena cava; RHV, right hepatic vein; CHV, central hepatic vein; LHV, left hepatic vein; RL, right lateral; CP, caudate process caudate lobe; RM, right medial; GB, gallbladder; Q, quadrate; LM, left medial; LL, left lateral; PP, papillary process caudate lobe; FL, fused lobe.

**Table 1 vetsci-09-00607-t001:** Number of tertiary portal (PV) and hepatic vein (HV) branches supplying each lobe.

	LM	LM	Q	PP	CP	RM	RL
**PV**	2(2); **3(3)**; 4(2)	**1(5)**; 2(1); 3(1)	1(1); **2(5)**	1(2); **2(5)**	2(2); **3(3)**; 4(1)	**1(6);** 4(1)	**1(4)**; 2(1); 3(1)
**HV**	2(1); **3(6)**	1(1); **2(3)**; **3(3)**	1(1); **2(4)**; 3(1)	0(1); **1(6)**	2(1); 3(1); **4(2)**; 5(1); 9(1)	**2(5)**; 3(2)	**3(4)**; 4(2)

LM, left medial; Q, quadrate; PP, papillary process; CP, caudate process; RM, right medial; RL, right lateral; HV, hepatic vein; PV, portal vein. Total tertiary branches supplying each liver lobe, with the most frequent pattern in bold. The number of specimens is in parenthesis ().

## Data Availability

The data that support the findings of this study are available from the corresponding author, S.V., upon reasonable request.

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
