# Peer review of "Preliminary Studies on the Intrahepatic Anatomy of the Venous Vasculature in Cats"

_vetsci, 2022, doi:10.3390/vetsci9110607_

Round 1

Reviewer 1 Report

This paper is well-written and describes the observed anatomy of the intrahepatic portal and hepatic venous systems using corrosion casted healthy feline livers. The images provided are very good at summarizing these findings. Livers from seven European Shorthair adult cats were harvested for use in this study, limiting conclusions relative to breed-specific patterns. It is assumed that the European Shorthair was chosen as the model for this study since it is the most common breed in Switzerland. This is also a natural cat breed that developed without selective breeding. You may consider providing a sentence or two as to why this particular breed was chosen as your study model and why the results may or may not be extrapolated to other cat breeds. A good argument was made as to why the findings of this study are relevant for small animal surgeons. Especially the noted major variation between dog and cat and the various branching patterns that can occur among cats.

Lines 9-11: Simple Summary: Consider changing to “The anatomy of the intrahepatic veins in cats has never been thoroughly described, thus veterinary surgeons have based their techniques on knowledge of canine liver anatomy for hepatic surgeries in both dogs and cats”.

Lines 11-13: The term “using” implies present tense, the term “described” is past tense. Tenses should match. Consider either: “We used corrosion cast...” OR “...cadavers, we describe the anatomy...”

Lines 18-22: Once again, tenses are in conflict. “The current study uses...”; “The results showed...”

Lines 22-23: “...as previously described in the literature.” Described specifically for cats, or is this relative to how it was described for dogs and inferred for cats?

Lines 90-93: It is recommended that cranial vs. caudal be referenced in these views.

Line 102: Missing closed parenthesis.

Lines 137-138, Figure 3: Since frequency “n/7” was provided for A1-A5, the same should be provided for “A”. “Most common portal pattern, n/7 specimens” OR “All specimens studied (n=7) exhibited a similar base pattern”.

Lines 159-160, Figure 4: Same comment as provided above relative to Figure 3.

Table 1: It may be better to use the same abbreviations that have been used throughout (i.e., LL, LM, Q, PP, CP, RM, RL) for column titles since these seem a bit crammed.

Reviewer 2 Report

The authors undertook to develop an interesting and very important from the clinical point of view issue, which is the venous vascularization of the cat's liver.

However, the presented conclusions should be treated as preliminary observations, due to the extremely low number of animals available to the researchers. This limitation is particularly important as regards the observed variants, which cannot be subjected to statistical analysis. Therefore, I strongly suggest changing the title to include the words "preliminary studies". The remaining comments relate to the anatomical nomenclature, which is inconsistent with the commonly used one (Nomina Anatomica Veterinaria). When describing venous vessels, it is customary to assume that the smaller vessels enter the larger ones, according to the blood flow. In the case of arterial vessels, the larger ones are divided into smaller branches. More serious objections concern liver lobation (lobus hapatis dexter lateralis, lobus hepatis dexter medialis, lobus quadratus, lobus caudatus, lobus hepatis sinister lateralis and lobus hepatis sinister medialis) and the names of the vessels. Vena portae is formed by fusion (or, as the authors prefer, divided into) ramus dexter and ramus sinister (and its pars transversa). You cannot use the terms right / left portal vein. The same goes for the hepatic veins. In addition, the authors use the term "central hepatic vein" instead of "v. hepatica media".

The above comments do not diminish the substantive value of the article, however, in its current form, in my opinion, it should not be published.

Please also correct a few fragments:

line 12: main portal and hepatic veins (there is one vena portae and three hepatic veins)

line 13: what does the term "main anatomy" mean

lines 73,74: the sentence is incomprehensible

line 186: "two common PV trunks" What common trunks are you talking about?
